# On Success and Simplicity:
# A Second Look at Transferable Targeted Attacks

**Zhengyu Zhao, Zhuoran Liu, Martha Larson**
Radboud University
{z.zhao,z.liu,m.larson}@cs.ru.nl

## Abstract

Achieving transferability of targeted attacks is reputed to be remarkably difficult. The current state of the art has resorted to resource-intensive solutions that necessitate training model(s) for each target class with additional data. In our investigation, we find, however, that simple transferable attacks which require neither model training nor additional data can achieve surprisingly strong targeted transferability. This insight has been overlooked until now, mainly because the widespread practice of attacking with only few iterations has largely limited the attack convergence to optimal targeted transferability. In particular, we, for the first time, identify that a very simple logit loss can largely surpass the commonly adopted cross-entropy loss, and yield even better results than the resource-intensive state of the art. Our analysis spans a variety of transfer scenarios, especially including three new, realistic scenarios: an ensemble transfer scenario with little model similarity, a worse-case scenario with low-ranked target classes, and also a real-world attack on the Google Cloud Vision API. Results in these new transfer scenarios demonstrate that the commonly adopted, easy scenarios cannot fully reveal the actual strength of different attacks and may cause misleading comparative results. We also show the usefulness of the simple logit loss for generating targeted universal adversarial perturbations in a data-free manner. Overall, the aim of our analysis is to inspire a more meaningful evaluation on targeted transferability. Code is available at `https://github.com/ZhengyuZhao/Targeted-Tansfer`.

## 1 Introduction

Deep neural networks have achieved remarkable performance in various machine learning tasks, but are known to be vulnerable to adversarial attacks [1]. A key property of adversarial attacks that makes them critical in realistic, black-box scenarios is their transferability [2, 3]. Current work on adversarial transferability has achieved great success for non-targeted attacks [4–12], while several initial attempts [3, 4, 13] at targeted transferability have shown its extreme difficulty. Targeted transferability is known to be much more challenging and worth exploring since it can raise more critical concerns by fooling models into predicting a chosen, highly dangerous target class.

However, so far state-of-the-art results can only be secured by *resource-intensive transferable attacks* [14–16]. Specifically, the FDA approach [14, 15] is based on modeling layer-wise feature distributions by training target-class-specific auxiliary classifiers on large-scale labeled data, and then optimizing adversarial perturbations using these auxiliary classifiers from across the deep feature space. The TTP approach [16] is based on training target-class-specific Generative Adversarial Networks (GANs) through global and local distribution matching, and then using the trained generator to directly generate perturbations on any given input image.

In this paper, we take a second, thorough look at current research on targeted transferability. Our main contribution is the finding that *simple transferable attacks* [4, 6, 8] that require neither model

training nor additional data can actually achieve surprisingly strong targeted transferability. We argue that this insight has been overlooked mainly because current research has unreasonably restricted the attack convergence by only using a small number of iterations (see detailed discussion in Section 3). Another key contribution of our work is, for the first time, demonstrating the general superiority of a very simple logit loss, which even outperforms the resource-intensive state of the art.

In order to validate the general effectiveness of simple transferable attacks, in Section 4.1, we conduct extensive experiments in a wide range of transfer scenarios. We test the commonly adopted single-model and ensemble transfer scenarios, but also introduce three new scenarios that are more challenging and realistic: an ensemble transfer scenario with little model similarity, a worse-case scenario with low-ranked target classes, and also a real-world attack on the Google Cloud Vision API. Experimental results in these new scenarios suggest that evaluation in only the commonly adopted, easy scenarios cannot reveal the actual strength of different attacks, and may cause misleading comparative results. Additional experiments in Section 4.2 have shown the better performance of the simple transferable attacks than the state-of-the-art resource-intensive approaches. Finally, in Section 4.3, inspired by the observation that the generated perturbations themselves reflect specific target semantics, we use the simple Logit attack to generate targeted Universal Adversarial Perturbations (UAPs) in a data-free manner. In contrast, recent advances in targeted UAPs [16–19] have inevitably relied on large-scale optimization over additional data.

Overall, we hope our analysis of the weakness of commonly adopted attack settings and transfer scenarios will inspire a more meaningful evaluation on targeted transferability.

## 2 Related Work

In this section, we review existing simple transferable attacks (Section 2.1), and also recent resource-intensive transferable attacks (Section 2.2). Finally, we discuss related work on generating universal adversarial perturbations.

### 2.1 Simple Transferable Attacks

We refer to transferable attacks that require neither model training nor additional data, but only use iterative optimization on a single (original) image as *simple transferable attacks*. Simple transferable attacks have been extensively studied in the non-targeted case [4–12], and also attempted in the targeted case [3, 4, 20]. These attacks are commonly built up on the well-known Iterative-Fast Gradient Sign Method (I-FGSM) [21, 22], which can be formulated as:

$$\boldsymbol{x}_0' = \boldsymbol{x}, \;\; \boldsymbol{x}_{i+1}' = \boldsymbol{x}_i' - \alpha \cdot \operatorname{sign}(\nabla_{\boldsymbol{x}} J(\boldsymbol{x}_i', y_t)), \tag{1}$$

where $\boldsymbol{x}_i'$ denotes the perturbed image in the $i$-th iteration, and $y_t$ is the target class label. In order to ensure the imperceptibility, the added perturbations are restricted with respect to some $L_p$ distance, i.e., satisfying $\|\boldsymbol{x}' - \boldsymbol{x}\|_p \leq \epsilon$. Current transferable attack methods have commonly adopted the $L_\infty$ distance, but can also be easily adapted to the $L_2$ distance based on an $L_2$ normalization [23].

For the loss function $J(\cdot, \cdot)$, most simple transferable attacks have adopted the Cross-Entropy (CE) loss. However, the CE has been recently shown to be insufficient in the targeted case due to its decreasing gradient problem [20]. To address this problem, the authors in [20] have proposed the **Po+Trip** loss, in which the Poincaré distance was used to adapt the gradients' magnitude:

$$L_{Po} = d(\boldsymbol{u}, \boldsymbol{v}) = \operatorname{arccosh}(1 + \delta(\boldsymbol{u}, \boldsymbol{v})),$$

$$\delta(\boldsymbol{u}, \boldsymbol{v}) = \frac{2 \cdot \|\boldsymbol{u} - \boldsymbol{v}\|_2^2}{(1 - \|\boldsymbol{u}\|_2^2)(1 - \|\boldsymbol{v}\|_2^2)}, \; \boldsymbol{u} = \frac{l(\boldsymbol{x}')}{\|l(\boldsymbol{x}')\|}, \; \boldsymbol{v} = \max\{\boldsymbol{v} - \xi, 0\}, \tag{2}$$

where $\boldsymbol{u}$ is the normalized logit vector and $\boldsymbol{v}$ is the one-hot vector with respect to the target class. $\xi = 10^{-5}$ is a small constant to ensure numerical stability. The following triplet loss is also integrated for pushing the image away from the original class while pulling it into the target class:

$$L_{Trip} = [D(l(\boldsymbol{x}'), y_t) - D(l(\boldsymbol{x}'), y_o) + \gamma]_+, \; D(l(\boldsymbol{x}'), y) = 1 - \frac{\|l(\boldsymbol{x}') \cdot y\|_1}{\|l(\boldsymbol{x}')\|_2 \|y\|_2}. \tag{3}$$

The overall loss function is then formulated as $L_{Po+Trip} = L_{Po} + \lambda L_{Trip}$. Note that in the original work, Po+Trip was evaluated in the commonly adopted, easy ensemble transfer scenario, which only involves models with similar architectures.

In addition to devising new loss functions, there are other transfer methods [4, 6, 8, 9] developed based on the assumption that preventing the attack optimization from overfitting to the specific source model can improve transferability. Such transfer methods can be easily plugged into different attacks without modifications, in contrast to the above methods that need to apply new attack loss functions. In this paper, we consider three [4, 6, 8] of such transfer methods that have been widely used in the literature, as described in the following text.

**Momentum Iterative-FGSM (MI-FGSM)** [4] integrates a momentum term, which accumulates previous gradients in order to achieve more stable update directions. It can be expressed as:

$$\boldsymbol{g}_{i+1} = \mu \cdot \boldsymbol{g}_i + \frac{\nabla_{\boldsymbol{x}} J(\boldsymbol{x}'_i, y_t)}{\|\nabla_{\boldsymbol{x}} J(\boldsymbol{x}'_i, y_t)\|_1}, \ \boldsymbol{x}'_{i+1} = \boldsymbol{x}'_i - \alpha \cdot \ \mathrm{sign}(\boldsymbol{g}_i), \tag{4}$$

where $\boldsymbol{g}_i$ is the accumulated gradients at the $i$-th iteration, and $\mu$ is a decay factor. Another similar technique that instead uses the Nesterov accelerated gradient was explored in [9].

**Translation Invariant-FGSM (TI-FGSM)** [6] randomly translates the input image during attack optimization in order to prevent the attack from overfitting to the specific source model. This approach is inspired by the data augmentation techniques used for preventing overfitting in normal model training. Instead of calculating gradients for multiple translated images separately, the authors have proposed an approximate solution to accelerate the implementation. It is achieved by directly computing locally smoothed gradients on the original image via convolution with a kernel:

$$\boldsymbol{x}'_{i+1} = \boldsymbol{x}'_i - \alpha \cdot \ \mathrm{sign}(\boldsymbol{W} * \nabla_{\boldsymbol{x}} J(\boldsymbol{x}'_i, y_t)), \tag{5}$$

where $\boldsymbol{W}$ is the convolution kernel used for smoothing. TI-FGSM was originally designed for boosting transferability with adversarially-trained models as target models and has been recently shown that a smaller kernel size should be used when transferring to normally-trained models [12].

**Diverse Input-FGSM (DI-FGSM)** [8] follows a similar idea to TI-FGSM, but applies random resizing and padding for data augmentation. Another important difference is that DI-FGSM randomizes augmentation parameters over iterations rather than fixing them as in TI-FGSM. The attack optimization of DI-FGSM can be formulated as:

$$\boldsymbol{x}'_{i+1} = \boldsymbol{x}'_i - \alpha \cdot \ \mathrm{sign}(\nabla_{\boldsymbol{x}} J(T(\boldsymbol{x}'_i, p), y_t)), \tag{6}$$

where the stochastic transformation $T(\boldsymbol{x}'_i, p)$ is implemented with probability $p$ at each iteration. In Section 4.1, we demonstrate that simple transfer attacks with these three transfer methods can actually achieve surprisingly strong targeted transferability in a wide range of transfer scenarios.

## 2.2 Resource-Intensive Transferable Attacks

Due to the broad consensus that achieving targeted transferability is extremely difficult, recent researchers have resorted to resource-intensive approaches that require training target-class-specific models on large-scale additional data. Specifically, the **Feature Distribution Attack (FDA)** [14] follows the same attack pipeline as the above simple transferable attacks, but requires auxiliary classifiers that have been trained on additional labeled data as part of the source model. Each auxiliary classifier is a small, binary, one-versus-all classifier trained for a specific target class at a specific layer. That is to say, the number of auxiliary classifiers is the number of layers that are probed multiplied by the number of target classes that are required to model [14]. The attack loss function of FDA can be formulated as:

$$L_{FDA} = J(\mathcal{F}_l(\boldsymbol{x}'), y_t) - \eta \frac{\|\mathcal{F}_l(\boldsymbol{x}') - \mathcal{F}_l(\boldsymbol{x})\|_2}{\|\mathcal{F}_l(\boldsymbol{x})\|_2}, \tag{7}$$

where each auxiliary classifiers $F_l(\cdot)$ can model the probability that a feature map at layer $l$ is from a specific target class $y_t$. **FDA$^{(N)}$+xent** [15] extends FDA by aggregating features from $L$ layers and also incorporating the cross-entropy loss $H(\cdot, \cdot)$ of the original network $\mathcal{F}(\cdot)$. The loss function of FDA$^{(N)}$+xent can be expressed as:

$$L_{FDA^{(N)}+xent} = \sum_{l \in L} \lambda_l (L_{FDA} + \gamma H(\mathcal{F}(\boldsymbol{x}'), y_t)), \ \text{where} \ \sum_{l \in L} \lambda_l = 1. \tag{8}$$

Very recently, **TTP** [16] has achieved state-of-the-art targeted transferability by directly generating perturbations using target-class-specific GANs that have been trained via matching the distributions of

perturbations and a specific target class both globally and locally. Specifically, the global distribution matching is achieved by minimizing the Kullback Leibler (KL) divergence, and the local distribution matching is by enforcing the neighbourhood similarity. In order to further boost the performance, data augmentation techniques, such as image rotation, crop resize, horizontal flip, color jittering and gray-scale transformation, have been applied during model training. We refer the readers to [16] for more technical details of TTP.

These two transferable attacks, FDA$^{(N)}$+xent and TTP, are resource intensive due to the use of large-scale model training and additional data. However, in Section 4.2, we show that simple transferable attacks, which require neither model training nor additional data, can actually achieve even better performance than them.

### 2.3 Universal Adversarial Perturbations

Previous research has shown the existence of Universal Adversarial Perturbations (UAPs), i.e., a single image perturbation vector that fools a classifier on multiple images [24]. UAPs have been extensively studied for non-targeted attacks [24–28], but also explored in the more challenging, targeted case [17–19]. Although recent studies have shown comparable performance of using reconstructed class impressions [25] or proxy datasets [18] to original training data, large-scale optimization over image data is still necessary for most existing methods. Differently, a data-free approach [26] has been proposed for non-targeted UAPs by iteratively optimizing randomly-initialized perturbations with an objective of disrupting the intermediate features of the model at multiple layers. However, this approach cannot be applied to targeted UAPs because targeted perturbations aim at a specific direction but not random disruption as in the non-targeted case. To bridge this gap, in Section 4.3, we demonstrate how the simple Logit attack can be used to generate targeted UAPs in a data-free manner.

## 3 New Insights into Simple Transferable Attacks

In this section, we revisit simple transferable targeted attacks, and provide new insights into them. Specifically, we demonstrate that simple transferable attacks that are based on existing transfer methods (TI-, MI-, and DI-FGSM) need more iterations to converge, and attacking with a simple logit loss can yield much better results than the commonly adopted Cross-Entropy (CE) loss.

### 3.1 Existing Transfer Methods with More Iterations Yield Good Results

Existing attempts have concluded that using simple transferable attacks to achieve targeted transferability is extremely difficult [3, 4, 13–15]. However, these attempts have been limited to the MI transfer method. Here, we tested all the three transfer methods. As can be seen form Figure 1, integrating all the three transfer methods leads to the best performance. In particular, we find that using only DI can actually yield substantial targeted transferability, while using only TI or MI makes little difference to the original poor targeted transferability. The fact that DI outperforms TI may be explained by the fact that DI randomizes the image augmentation parameters over iterations rather than fixing them as in TI. In this way, the gradients towards the target class become more generic and so avoid overfitting to the white-box source model. MI is essentially different from DI and TI because it can only stabilize update directions but not serve to achieve more accurate gradient directions towards a specific (target) class.

As we have pointed out in Section 1, common practice of generating transferable targeted perturbations [13–15, 20] has limited the attack optimization to few iterations (typically $\leq 20$). This is somewhat understandable given that extensive research on non-targeted transferability has done the same. However, as can be seen from Figure 1, targeted attacks actually require much more iterations to converge to optimal transferability, in contrast to the fast convergence of non-targeted attacks. This implies that evaluating the targeted transferability under only few iterations is problematic. On the one hand, comparing different optimization processes that have not converged is not meaningful and may cause misleading comparisons (see evidence in Section 4.1). This observation is consistent with the evaluation suggestion in [29] that restricting the number of iterations without verifying the attack convergence is one of the common pitfalls in evaluating adversarial robustness. Several advanced defenses have been defeated by simply increasing the number of iterations [30]. On the other hand,

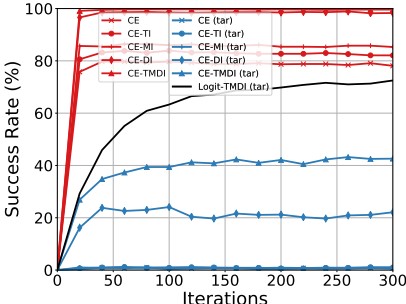
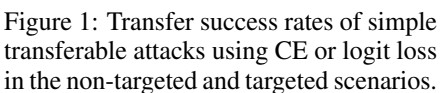

Figure 1: Transfer success rates of simple transferable attacks using CE or logit loss in the non-targeted and targeted scenarios.

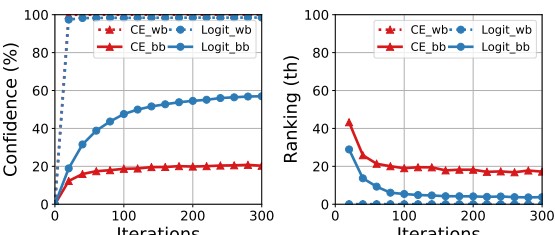

Figure 2: White-box (wb) and black-box (bb) attack performance in terms of the predicted confidence (left, higher is better) and ranking (right, lower is better) of the target class.

considering the realistic threat model, it is not meaningful to artificially restrict the computational power of a practical attack (e.g., to fewer than several thousand attack iterations) [31].

## 3.2 A Simple yet Strong Logit Attack

Existing simple transferable attacks have commonly adopted the Cross-Entropy (CE) loss. However, as pointed out in [20], during the attack optimization, the CE loss will cause the gradient to decrease and tend to vanish as the number of iterations is increased. To address this problem, the Po+Trip loss [20] takes a very aggressive strategy by arbitrarily reversing the decrease of the gradient, i.e., gradually increasing the magnitude of the gradients over iterations. However, we argue that this operation has led to too large step size, and as a result cause the attack optimization to overshoot the minima. Our results in Section 4.1 support this argument by showing that Po+Trip even yielded worse results than CE in the ensemble transfer scenario with diverse model architectures, since the loss surface is relatively non-smooth.

Here, for the loss function, we eliminate the final softmax function used in the CE loss and just backpropagate the gradients from the logit output:

$$L_{Logit} = -l_t(\boldsymbol{x}'), \tag{9}$$

where $l_t(\cdot)$ denotes the logit output with respect to the target class. Although the idea of attacking logits is not new, its superior performance in targeted transferability has not been recognized so far. We also find that using the well-known logit-based loss, C&W [32], yields consistently worse results (see detailed comparisons in Appendix A). Another logit loss that is similar to the C&W loss has also been adopted by [18], but in the task of generating UAPs with large-scale data.

Below, we show that this logit loss leads to stronger gradients than the CE loss. As can be observed from Equation 10, the gradient of the CE loss with respect to the target logit input, $z_t$, will monotonically decrease as the probability of the target class, $p_t$, increases during attack optimization. In addition, due to the use of the softmax function, $p_t$ will quickly reach 1, and as a result the gradient tends to vanish. This phenomenon makes the attack hard to improve even with more iterations applied. Differently, as shown by Equation 11, the gradient of the logit loss equals a constant. In this way, the attack can keep improving as the number of iterations is increased. In Appendix B, we provide further comparisons on the trends of loss/gradient magnitude and the target logit value over iterations, which show that the logit loss leads to better results than both CE and Po+Trip.

$$L_{CE} = -1 \cdot \log(p_t) = -\log(\frac{e^{z_t}}{\sum e^{z_j}}) = -z_t + \log(\sum e^{z_j}),$$
$$\frac{\partial L_{CE}}{\partial z_t} = -1 + \frac{\partial \log(\sum e^{z_j})}{\partial e^{z_t}} \cdot \frac{\partial e^{z_t}}{\partial z_t} = -1 + \frac{e^{z_t}}{\sum e^{z_j}} = -1 + p_t. \tag{10}$$

$$L_{Logit} = -z_t, \quad \frac{\partial L_{Logit}}{\partial z_t} = -1. \tag{11}$$

Table 1: Targeted transfer success rates (%) in the single-model transfer scenario. We consider three attacks with different loss functions: cross-entropy (CE), Poincaré distance with Triplet loss (Po+Trip) [20], and the logit loss. Results with 20/100/300 iterations are reported.

| Attack | Source Model: Res50 | | | Source Model: Dense121 | | |
|---|---|---|---|---|---|---|
| | →Dense121 | →VGG16 | →Inc-v3 | →Res50 | →VGG16 | →Inc-v3 |
| CE | 26.9/39.4/42.6 | 17.3/27.3/30.4 | 2.4/3.8/4.1 | 13.1/17.3/19.4 | 7.7/10.8/10.9 | 1.9/3.3/3.5 |
| Po+Trip | 26.7/53.0/54.7 | 18.8/34.2/34.4 | 2.9/6.0/5.9 | 10.1/14.7/14.7 | 6.7/8.3/7.7 | 2.1/3.0/2.7 |
| Logit | **29.3/63.3/72.5** | **24.0/55.7/62.7** | **3.0/7.2/9.4** | **17.2/39.7/43.7** | **13.5/35.3/38.7** | **2.7/6.9/7.6** |

| Attack | Source Model: VGG16 | | | Source Model: Inc-v3 | | |
|---|---|---|---|---|---|---|
| | →Res50 | →Dense121 | →Inc-v3 | →Res50 | →Dense121 | →VGG16 |
| CE | 0.7/0.4/0.6 | 0.5/0.3/0.1 | 0/0.1/0 | 0.6/**2.1**/2.4 | 0.8/2.5/2.9 | 0.7/1.6/2.0 |
| Po+Trip | 0.6/0.8/0.5 | 0.6/0.6/0.7 | 0.2/0.1/0.1 | 0.6/2.0/2.5 | 0.8/**3.1**/3.3 | 0.5/2.1/2.0 |
| Logit | **3.3/8.7/11.2** | **3.6/11.7/13.2** | **0.2/0.7/0.9** | **0.8**/1.6/**2.9** | **1.2**/2.8/**5.3** | **0.7/2.2/3.7** |

## 4 Experimental Evidence on Simple Transferable Attacks

In this section, we provide experimental evidence to show the general effectiveness of simple transferable attacks. Firstly, in Section 4.1, we evaluate the simple transferable attacks in a variety of transfer scenarios, including single-model transfer, ensemble transfer (easy and challenging scenarios), a worse-case scenario with low-ranked target classes, and a real-world attack on the Google Cloud Vision API. Then, in Section 4.2, we compare the simple transferable attacks with two state-of-the-art resource-intensive transferable attacks, FDA$^{(N)}$+xent [15] and TTP [16]. Finally, in Section 4.3, we apply the Logit attack to achieving targeted UAPs in a data-free manner.

Following recent work [14–16, 20], we focus on targeted transferability of ImageNet-like images, which is known to be much more difficult than other data sets (e.g, MNIST and CIFAR-10) with smaller-size images and fewer classes. Specifically, we used the 1000 images from the development set of the ImageNet-Compatible Dataset[1], which was introduced along with the NIPS 2017 Competition on Adversarial Attacks and Defenses. All these images are associated with 1000 ImageNet class labels and cropped to 299×299 before use. Our experiments were run on an NVIDIA Tesla P100 GPU with 12GB of memory.

### 4.1 Simple Transferable Attacks in Various Transfer Scenarios

We tested three different attack losses: CE, Po+Trip [20] and Logit. All attacks used TI, MI, and DI with optimal hyperparameters provided in their original work. Specifically, $\|\boldsymbol{W}\|_1 = 5$ was used for 'TI' as suggested by [12]. For each image, we used the target label that was officially specified in the dataset. If not mentioned specifically, all attacks were run with 300 iterations to ensure convergence. When being executed with a batch size of 20, the optimization process took about three seconds per image. A moderate step size of 2 was used for all attacks, and the results were shown to be not sensitive to the setting of step size (see evidence in Appendix C). We considered four diverse classifier architectures: ResNet [33], DenseNet [34], VGGNet [35], and Inception [36]. Following the common practice, the perturbations were restricted by $L_\infty$ norm with $\epsilon = 16$.

**Single-model transfer.** Table 1 reports the targeted transferability when transferring between each pair of different model architectures. As can be seen, the logit loss outperformed CE and Po+Trip by a large margin in almost all cases. When comparing different model architectures, we can find that the attacks achieved lower performance when transferring from the VGGNet16 or Inception-v3 than from ResNet50 or DenseNet121. This is consistent with the observations in [14, 15] and may be explained by the fact that skip connections in ResNet50 and DenseNet121 boosts transferability [37]. Another finding is that when using Inception-v3 as the target model, the transfer success rates were always low. This might be explained by the heavily engineered nature of the Inception architecture, i.e., the Inception architecture has multiple-size convolution and two auxiliary classifiers.

---

[1]https://github.com/cleverhans-lab/cleverhans/tree/master/cleverhans_v3.1.0/examples/nips17_adversarial_competition/dataset.

Table 2: Targeted transfer success rates (%) in the commonly adopted, easy ensemble transfer scenario, where the hold-out target model (denoted by '-') and the ensemble models share similar architectures. Results with 20/100 iterations are reported.

| Attack | -Inc-v3 | -Inc-v4 | -IncRes-v2 | -Res50 | -Res101 | -Res152 | Average |
|--------|---------|---------|------------|--------|---------|---------|---------|
| CE | 48.8/85.3 | 47.2/83.3 | 47.5/83.9 | 50.9/89.8 | 58.5/**93.2** | 56.7/90.7 | 51.6/87.7 |
| Po+Trip | **59.3**/84.4 | **55.0**/82.4 | 51.4/80.8 | 56.9/85.0 | 60.5/87.9 | 57.6/85.7 | 56.8/84.4 |
| Logit | 56.4/**85.5** | 52.9/**85.8** | **54.4/85.1** | **57.5/90.0** | **64.4**/91.4 | **61.3/90.8** | **57.8/88.1** |

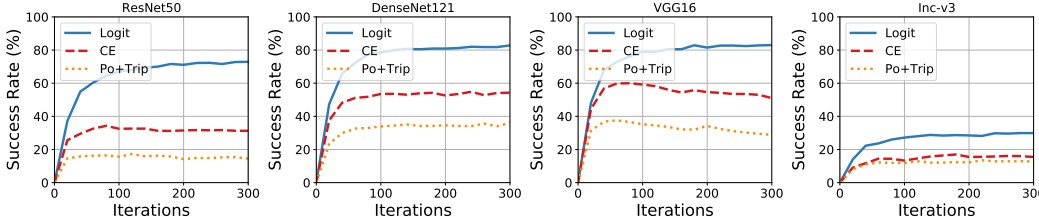

Figure 3: Targeted transfer success rates (%) in our challenging ensemble transfer scenario, where each hold-out target model shares no similar architecture with the source models used for ensemble.

**Ensemble transfer in both easy and challenging scenarios.** A common approach to further boosting transferability is to generate perturbations on an ensemble of white-box source models. Following the common practice, we simply assigned equal weights to all the source models. We first look at the commonly adopted ensemble transfer scenario [4, 6, 20, 38] in which each hold-out target model shares a similar architecture with some of the white-box ensemble models. As can be seen from Table 2, the transfer success rates of all three attacks have got saturated when given enough iterations to converge. As a result, this transfer scenario could not fully reveal the actual strength of different attacks. We can also observe that Po+Trip performed better than the CE loss only when the attack optimization is unreasonably restricted to 20 iterations, but became even worse with enough iterations. This finding suggests that evaluating different attacks under only few iterations may cause misleading comparative results.

We next considered a more challenging transfer scenario with no architectural overlap between the source ensemble models and the target model, in order to fully reveal the potential of different attacks. This scenario is also more realistic since it is hard for an attacker to know the specific architecture of a real-world target mode. Figure 3 shows that in this scenario, the Logit largely outperformed CE and Po+Trip. In addition, the results of both CE and Logit were substantially improved over the single-model transfer results reported in Table 1. However, Po+Trip performed even worst in some cases maybe because its use of arbitrarily increasing the gradient magnitude has caused the optimization to overshoot the minima in this ensemble transfer scenario where the loss surface is relatively non-smooth due to the model diversity. Note that as in the single transfer scenario, transferring to Inception-v3 is still the most difficult.

**A worse-case transfer scenario with low-ranked target classes.** In conventional security studies, a comprehensive evaluation commonly involves a range of attack scenarios with varied difficulty. Existing work on white-box adversarial attacks [32, 21, 23, 38] has also looked at different cases with varied difficulty regarding the ranking position of the target class in the prediction list of the original image.

Table 3: Targeted transfer success rates (%) when varying the target from the high-ranked class to low.

| Attack | 2nd | 10th | 200th | 500th | 800th | 1000th |
|--------|-----|------|-------|-------|-------|--------|
| CE | **89.9** | 76.7 | 49.7 | 43.1 | 37.0 | 25.1 |
| Po+Trip | 82.6 | 77.6 | 58.4 | 53.6 | 49.1 | 38.2 |
| Logit | 83.8 | **81.3** | **75.0** | **71.0** | **65.1** | **52.8** |

Specifically, in the best case, the targeted success is basically equal to non-targeted success, i.e., an attack is regarded to be successful as long as it can succeed on any arbitrary target other than the original class. In the average case, the target class is randomly specified, while in the worst case, the target is specified as the lowest-ranked/least-likely class.

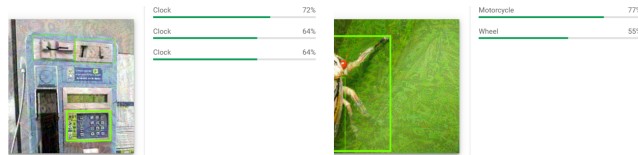

Target class: "analog clock"          Target class: "mountain bike"

Figure 4: Successful targeted adversarial images on Google Cloud Vision generated by the Logit attack with ensemble transfer. More examples can be found in Appendix D.

Table 4: Non-targeted and targeted transfer success rates (%) of different attacks on Google Cloud Vision.

|  | CE | Po+Trip | Logit |
|---|---|---|---|
| Targeted | 7 | 8 | **18** |
| Non-targeted | **51** | 44 | **51** |

However, to the best of our knowledge, current evaluation of transfer-based attacks has been limited to the best and average cases. To address this limitation, we consider a worse-case transfer scenario by varying the target from the highest-ranked class gradually to the lowest one. As can be seen from Table 3, there exists a non-negligible correlation between the ranking position of the target class and the targeted transferability. More specifically, it becomes increasingly difficult as the target moves down the prediction list. We can also observe that the results with higher-ranked targets might not reveal the actual strength of different attacks as in the more realistic, worse cases with lower-ranked targets. In particular, only looking the best case with the highest-ranked target may lead to a misleading conclusion that CE leads to the most effective attack. This finding suggests that a more meaningful evaluation on targeted transferability should further increase difficulty beyond the current best and average cases.

**Transfer-based attacks on Google Cloud Vision.** Most existing work on fooling real-world computer vision systems has been focused on the query-based attacks, where a large number of queries are required [39–41]. Although several recent studies have also explored real-world transfer-based attacks, they were limited to face recognition and the non-targeted attacks [42–44]. In contrast, we applied the simple transferable attacks in the more challenging, targeted case on a more generally-used image recognition system, the Google Cloud Vision API. Specifically, we used the targeted adversarial images generated on the ensemble of all four diverse source models with 300 iterations.

The API predicts a list of semantic labels along with confidence scores. Specifically, only the top classes with confidence no lower than $50\%$ are returned, and at most 10 classes are shown. Note that the confidence score here is not a probability (which would sum to one). We measured both the targeted and non-targeted transferability. Since all returned labels are with relatively high confidence ($\geq 50\%$), we do not limit our measure of success rates to only top-1 class. Instead, for non-targeted success, we measured whether or not the ground-truth class appeared in the returned list, while for targeted success, whether or not the target class appeared. Due to the fact that the semantic label set predicted by the API does not exactly correspond to the 1000 ImageNet classes, we treated semantically similar classes as the same class.

Table 4 reports the results averaged over 100 images that originally yield correct predictions. As can be seen, in general, achieving targeted transfer success is much more difficult than non-targeted success. In particular, the Logit attack achieved the best targeted transferability, with quasi-imperceptible perturbations shown in Figure 4. Our results reveal the potential vulnerability of Google Cloud Vision against simple transfer-based attacks, which require no query interaction.

## 4.2 Simple vs. Resource-Intensive Transferable Attacks

In this subsection, we compared simple transferable attacks with state-of-the-art resource-intensive approaches, TTP [16] and FDA$^{(N)}$+xent [15], which necessitate training target-class-specific models on additional data.

**Compared with TTP.** We compared the Logit attack with the state-of-the-art TTP, which is based on training target-class-specific GANs on additional data. We tested both Logit and TTP on our dataset following the "10-Targets (all-source)" setting in [16]. We chose ResNet50 as the white-box model in the single-model transfer scenario and an ensemble of ResNet{18,50,101,152} in the ensemble transfer scenario. DenseNet121 and VGG16_bn are tested as the target models. Note that the same knowledge of the white-box model is available to both attacks but it is leveraged in different ways.

Specifically, for the Logit attack, the white-box model is used as a source model for iterative attack optimization, while for TTP, it is used as a discriminator during training the GANs.

As shown in Table 5, under the commonly adopted $\epsilon = 16$, the Logit attack can achieve comparable results to TTP in all cases. Specifically, we can observe that the model ensemble is more helpful to Logit than for TTP. This might be because even with the single model as the discriminator, TTP can learn good enough features of target semantics by training with the objective of matching the perturbation and target class distributions with large-scale data. The clearer target semantics learned by TTP can be confirmed by comparing the unbounded pertur-

Table 5: Targeted transfer success rates (%) of Logit vs. TTP in single-model and ensemble transfer scenarios under two norm bounds.

| Bound | Attack | D121 | V16 | D121-ens | V16-ens |
|---|---|---|---|---|---|
| $\epsilon = 16$ | TTP | **79.6** | **78.6** | 92.9 | 89.6 |
| | Logit | 75.9 | 72.5 | **99.4** | **97.7** |
| $\epsilon = 8$ | TTP | 37.5 | 46.7 | 63.2 | 66.2 |
| | Logit | **44.5** | **46.8** | **92.6** | **87.0** |

bations achieved by TTP (e.g., Figure 3 in [16]) with those by the Logit shown in Figure 5.

This fact that TTP perturbations heavily rely on semantic patterns of the target class might cause TTP to degrade under lower norm bounds. To validate this assumption, we further compared Logit and TTP under $\epsilon = 8$. As expected, the Logit attack consistently surpassed TTP, especially with a very large margin in the ensemble transfer scenario. The different comparing results for the two perturbation sizes also suggest that comparing attacks only under a single perturbation size may not reveal their characteristics.

Table 6: Targeted transfer success rates (%) of unbounded adversarial images by different attacks with the same iteration budget.

|  | FDA$^{(4)}$ +xent | CE | Po+Trip | Logit |
|---|---|---|---|---|
| Res50→Dense121 | 65.8 | 69.3 | **88.1** | 84.1 |
| Res50→VGG16 | 48.1 | 54.1 | 67.8 | **74.2** |

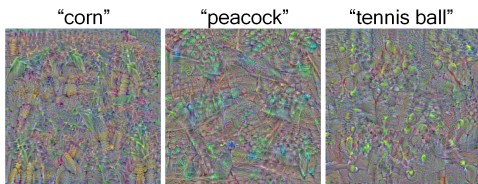

"corn"    "peacock"    "tennis ball"

Figure 5: Unbounded adversarial examples that reflect target semantics. More examples can be found in Appendix E.

**Compared with FDA$^{(N)}$+xent.** We compared the three simple transferable attacks (CE, Po+Trip, and Logit) with FDA$^{(N)}$+xent on generating unbounded adversarial images, in consistence with "distal transfer" in [15][2]. Specifically, the adversarial perturbations were initialized as random Gaussian noise and allowed to be as large as possible. Although such unbounded adversarial images may not be practically compelling, they can provide better isolated indication on transferability by eliminating the dependency on the source images and the bound restrictions. The results were averaged over 4000 image examples, each of which was optimized towards a random target class. As suggested by [15], the MI transfer method was removed since it empirically harms the performance in this unbounded case.

Table 6 shows that all the three simple transferable attacks achieved stronger targeted transferability than FDA$^{(N)}$+xent. As can be seen from Figure 5, the unbounded adversarial perturbations can somehow reflect the target semantics. This finding suggests that achieving targeted transferability relies on robust, semantic features [45] that are expected to be learned by various models and also understood by humans. In this way, achieving targeted transferability is fundamentally different from non-targeted transferability, for which attacking non-robust features is known to be sufficient [45]. It is also worth noting that in practical scenarios with small norm bounds, the semantically-aligned perturbations would not be expected to change human judgements.

### 4.3  Simple Logit Attack for Targeted UAPs in a Data-Free Manner

The above observation that the perturbations can reflect certain target semantics motivates us to apply the Logit attack to achieving targeted Universal Adversarial Perturbations (UAPs), which can drive multiple original images into a specific target class. Existing attempts at achieving targeted UAPs have mainly relied on large-scale optimization over additional data [17–19]. However, the

---

[2]We conducted comparisons only in this setting since the authors have not (yet) released their source code.

Table 7: Success rates (%) of targeted UAPs generated by CE and Logit attacks for different models.

| Attack | Inc-v3 | Res50 | Dense121 | VGG16 |
|--------|--------|-------|----------|-------|
| CE | 2.6 | 9.2 | 8.7 | 20.1 |
| Logit | **4.7** | **22.8** | **21.8** | **65.9** |



Figure 6: UAPs ($\epsilon = 16$, VGG16) with different classes using CE and Logit. More examples can be found in Appendix F.

simple Logit attack can be easily extended to generate targeted UAPs in a data-free manner. The only difference from the above transferable Logit attack is that here a mean image (all pixel values set as 0.5 out of [0,1]) is used as the original image.

In our experiment, for each target class, we generated a single targeted UAP vector ($\epsilon = 16$) with 300 iterations and applied it to all 1000 images in our dataset. Table 7 reports the results averaged over all the 1000 ImageNet classes. As can be seen, the logit loss can yield substantial success, remarkably outperforming the CE loss. This can be confirmed by Figure 6, which shows the Logit attack can yield more semantically-aligned perturbations than CE. This observation also supports the claim from [18] that universal perturbations contain dominant features, and images act like noise with respect to perturbations.

## 5    Conclusion and Outlook

In this paper, we have demonstrated that achieving targeted transferability is not as difficult as current work concludes. Specifically, we find that simple transferable attacks can actually achieve surprisingly strong targeted transferability when given enough iterations for convergence. We have validated the effectiveness of simple transferable attacks in a wide range of transfer scenarios, including three newly-introduced challenging scenarios. These challenging scenarios have better revealed the actual strength of different attacks. In particular, we demonstrate that a very simple Logit attack is superior in all transfer scenarios, achieving even better results than the state-of-the-art resource-intensive approaches. We also show the potential usefulness of the Logit attack for generating targeted universal adversarial perturbations in a data-free manner. Overall, we hope our findings will inspire future research to conduct a more meaningful evaluation on targeted transferability. Our future work will focus on studying why different model architectures yield different transferability. In particular, the very low success rates when targeting Inception-v3 should be explored. Moving forward, there needs to be a more comprehensive discussion on the resource consumption of different attacks from multiple aspects, such as training and inference time, hardware resources, and data size.

Strong transferability can obviously benefit black-box applications of adversarial images for social good, such as protecting user privacy [42, 43, 46–48]. In addition, it will also motivate the community to design stronger defenses given our finding that even simple attacks can generate highly transferable adversarial images. It remains a possibility that our methodology may be misused by malicious actors to break legitimate systems. However, we firmly believe that the help that our paper can provide to researchers significantly outweighs the help that it may provide an actual malicious actor.

## Acknowledgments

This work was carried out on the Dutch national e-infrastructure with the support of SURF Cooperative.

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
