# Appendix of "On Success and Simplicity: A Second Look at Transferable Targeted Attacks"

## A    Logit Loss Vs. C&W Loss

We compared the logit loss to C&W loss, which is also based on directly optimizing logits, with varied settings of the confidence controlling parameter $K$ [1]. As can be seen from the Figure 1, the logit loss consistently outperformed C&W by a large margin. Another interesting finding is that C&W converged slowly, which caused it to perform even worse than CE at an early stage. We can also observe that increasing $K$ generally improved C&W but after a certain point, it yielded even worse results. Directly using C&W without the $K$ yielded similar good performance as that of other proper settings of $K$. The consistently inferior performance of C&W may be because the C&W loss also involves suppressing other classes, which is not necessary here and could stand in the way of making the target logit as high as possible.

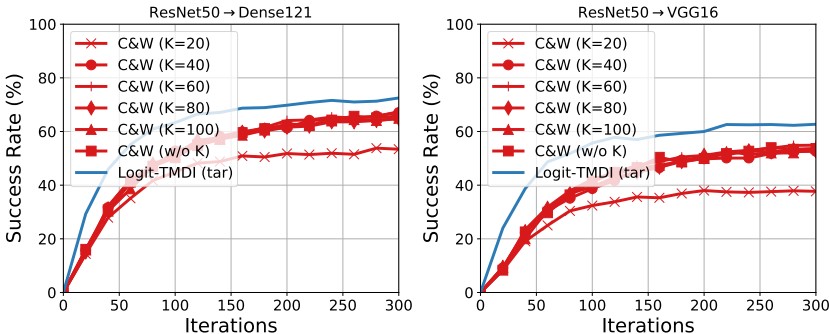

Figure 1: Targeted transfer success rates (%) of the logit loss compared to the C&W loss varied settings of the confidence controlling parameter $K$ [1] in the single-model transfer scenario.

## B    Theoretical Analysis of Different Losses

Following related work [2, 3], we plot the loss and gradient trends, and also the target logit trend over iterations in order to demonstrate why the logit loss can address the decreasing gradient problem of the CE loss and why it can address the problem better than the Po+Trip loss proposed in [2]. Specifically, for normalization, we re-scale all the losses/gradients by dividing their values of the first iteration. We use the $L_1$ (Taxicab) norm to represent the gradient magnitude. As can be observed by Figure 2, the loss value of CE quickly approaches zero and the Po+Trip loss stays almost unchanged with a small oscillation. In contrast, the logit loss can be minimized continuously over iterations with relatively large gradients. Overall, the use of the logit loss makes sure that the target logit can be continuously increased over iterations and finally reaches a very high value.

35th Conference on Neural Information Processing Systems (NeurIPS 2021).

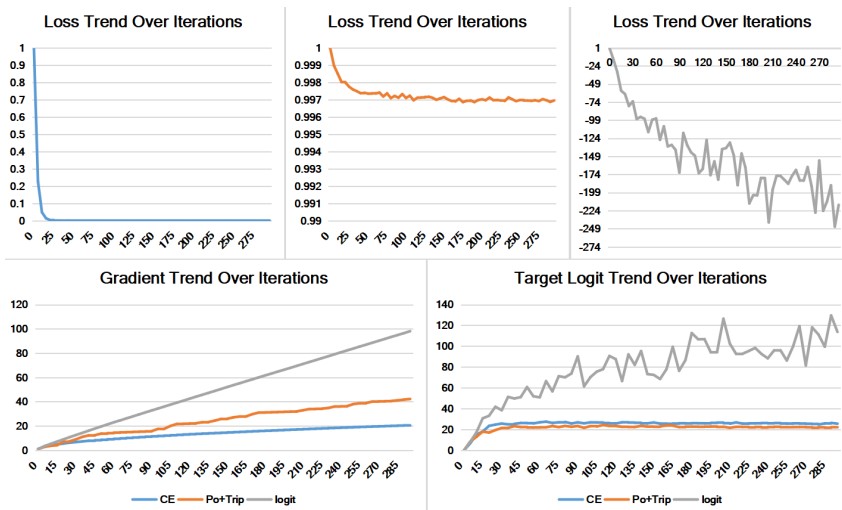

Figure 2: The loss, gradient, and target logit trends over iterations of attacks with three different losses in the Res50→Dense121 transfer scenario. The results are averaged over 100 images randomly selected from our dataset.

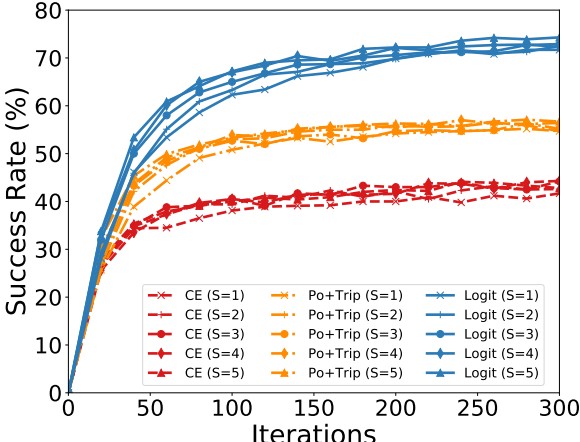

Figure 3: Targeted transferability of three simple transferable attacks (CE-TMDI, Po+Trip-TMDI and Logit-TMDI) with varied step size, S.

## C Attacks with Varied Step Sizes

Recent work has shown that enlarging the step size can improve non-targeted transferability since it can help attack optimization escape from poor local optima [4]. Here we also explore the impact of step size setting on targeted transferability. As can be seen from Figure 3, in general, all attacks are not sensitive to the change of step size, with only a slight improvement when using a larger step size. We can also observe that the logit attack consistently outperforms the other two in all cases.

# D    Adversarial Images on Google Cloud Vision

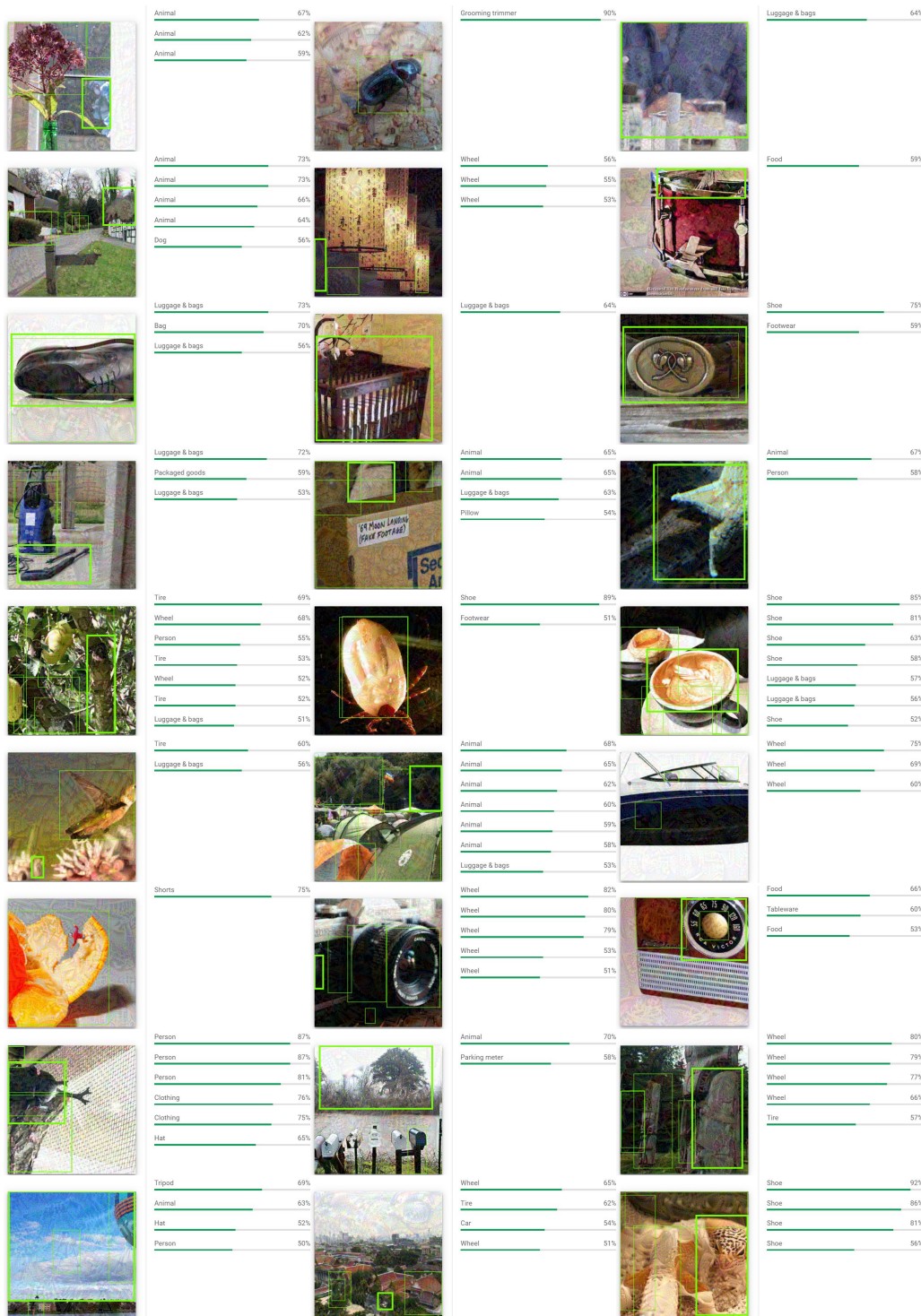

Figure 4: Adversarial images for attacking Google Cloud Vision.

# E   Unbounded Adversarial Images

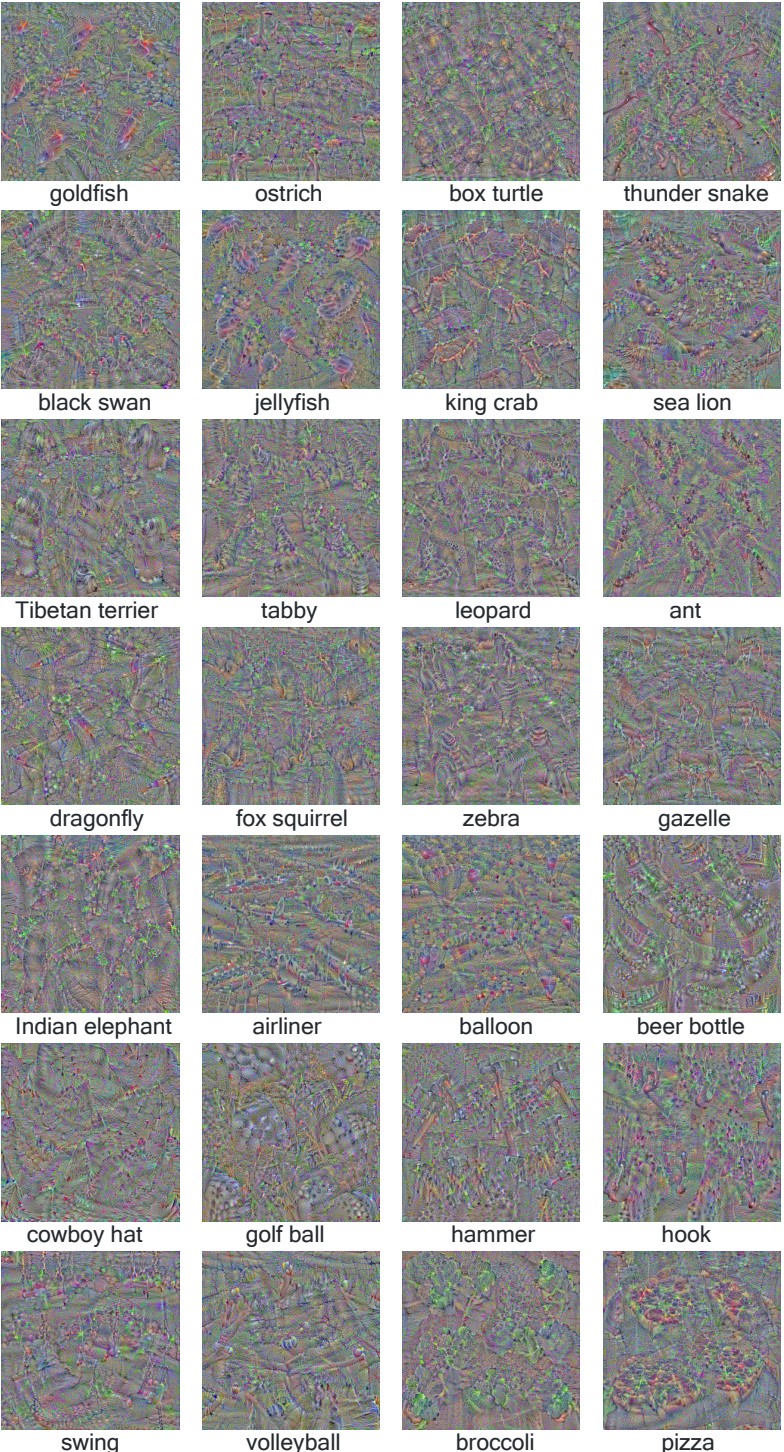

Figure 5: Unbounded adversarial images targeting different classes.

## F   Targeted Universal Adversarial Perturbations (UAPs)

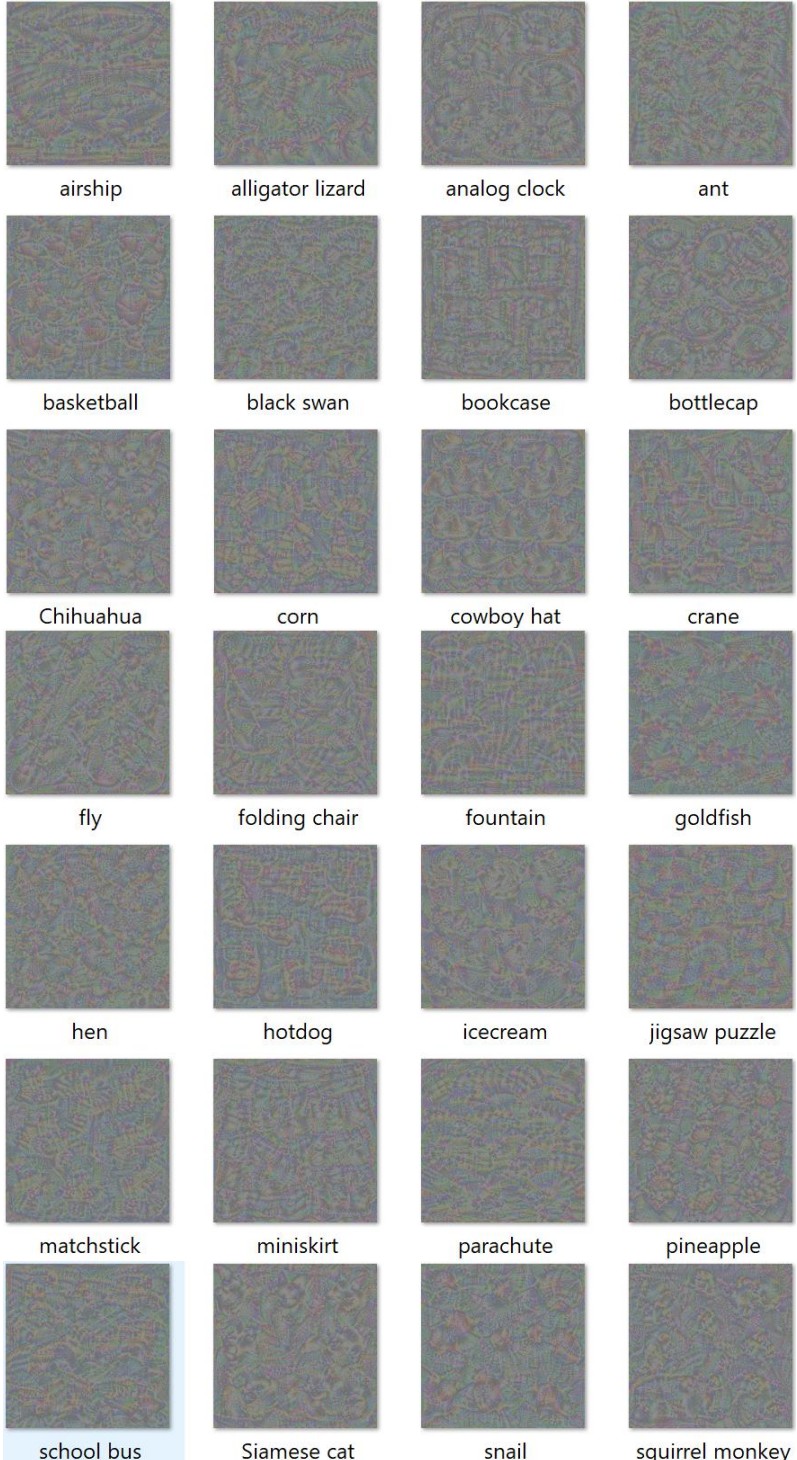

Figure 6: Targeted universal adversarial perturbations ($\epsilon = 16$) for different classes.