# OpenReview forum: "On Success and Simplicity: A Second Look at Transferable Targeted Attacks"
_NeurIPS.cc/2021/Conference — NeurIPS 2021 Poster_

### Official Review · Reviewer_c8VG · 2021-07-06

**Rating:** 6
**Confidence:** 2

**Summary:**

The paper provides new insights for transfer-based targeted attacks. The paper describes that target attacks with simple logit loss archive high transferability under sufficient iterations. Various experiments with realistic settings demonstrate the transferability of attacks with the logit loss. In addition, the paper shows the usefulness of the logit loss for generating universal adversarial perturbations.

**Limitations And Societal Impact:**

As current limitations, the authors have addressed that the paper does not explore what kind of model properties causes results.

The authors have described societal impacts of their attacks.

**Main Review:**

Pros:
- The use of logit loss is simple but very effective.
- The paper provides various experiments with realistic settings.

Cons:
- There is no theoretical result to support empirical findings.
- Some experimental settings are not clear.

The idea in the paper is very simple but shows strong empirical results. Although the paper does not provide theoretical insights, the findings with various experiments are useful for researchers in this field.

Some comments about Cons are described bellow:

- In line 192, the discussion about C&W loss seems not sufficient, as the original C&W loss has a hyperparameter (i.e., $\kappa$ in the original C&W paper) and there are some variants of C&W loss. Since the value of the hyperparameter would strongly affect the performance in this setting, the paper should describe the definition of the used C&W loss precisely (including the hyperparameter). Without the precise definition, it is difficult to evaluate whether the comparison between logit loss and C&W loss is properly held.

- In the experiments for Table 2, Figure 3, and Table3, the paper does not provide precise detailed information for ensemble source models. There is some information, but not sufficient. Since the paper adopts various settings for various experiments, the settings should be accurately described.

minor comments:

- In Figure 1, although there is only the result for logit-TMDI, the results for other settings (e.g., Logit-MI) would be useful.

- In Section 4.3, it is not clear whether the UAP vector is bounded or unbounded.


**Time Spent Reviewing:**

5 hours

---

> ### Author Response · Authors · 2021-08-10
> **Response to Reviewer c8VG**
>
> **Q1:** Theoretical analysis to explain the superiority of the logit attack.
>
> **A1:** Please see our response to Q1 of the Reviewer yhEq as shown above.
>
> **Q2:** Some experimental settings about C&W and source models for ensemble transfer are not clear.
>
> **A2:** As can be seen from the following table, the logit loss consistently outperforms C&W by a large margin. Another interesting finding is that C&W converges slowly, which causes it to perform even worse than CE at an early stage. We can also observe that increasing K generally improves C&W but after a certain point, it yields even worse results. Directly using C&W without the K (as we have adopted in our original paper) yields similar good performance as that of other proper settings of K. We will include these findings in our revised paper.
>
> | $\epsilon$=16/255|    Res50->D121 	|   Res50->VGG16 	|
> |-----------:	|---------------:	|---------------:	|
> |  C&W(K=20) 	| 14.4/44.9/53.4 	|  8.3/32.4/37.7 	|
> |  C&W(K=40) 	| 15.8/52.8/67.1 	|  9.2/38.7/52.6 	|
> |  C&W(K=60) 	| 14.3/51.8/67.4 	|  8.8/42.2/54.9 	|
> |  C&W(K=80) 	| 15.1/51.3/64.8 	|  9.4/39.4/52.8 	|
> | C&W(K=100) 	| 15.8/51.3/64.9 	|  9.4/42.4/53.8 	|
> | C&W(w/o K) 	| 16.1/50.2/65.8 	|  8.3/41.2/53.7 	|
> |         CE 	| 26.9/39.4/42.6 	| 17.3/27.3/30.4 	|
> |    Po+Trip 	| 26.7/53.0/54.7 	| 18.8/34.2/34.4 	|
> |      Logit 	| **29.3/63.3/72.5**	| **24.0/55.7/62.7** 	|
>
> As stated in our paper, for the source models in ensemble transfer with similar model architectures, we have adopted the exact same settings as in previous work [1-4]. In the revised version, we will describe them in more detail. For the ensemble transfer with little model similarity, we use the four models we have introduced in the first paragraph of Section 4.1.
>
> [1] Tramèr et al., Ensemble adversarial training: Attacks and defenses, ICLR 2018.
>
> [2] Dong et al., Boosting adversarial attacks with momentum, CVPR 2018.
>
> [3] Dong et al., Evading defenses to transferable adversarial examples by translation-invariant attacks, CVPR 2019.
>
> [4] Li et al., Towards transferable targeted attack, CVPR 2020.
>
> We will also address the minor comments on logit attack with other settings of -TMDI and UAP bound in the revised paper.

---

### Official Review · Reviewer_nyxN · 2021-07-14

**Rating:** 7
**Confidence:** 3

**Summary:**

This paper proposed a very interesting observation that the simple logit attack can consistently achieve higher targeted transferability than SOTA transferable attacks. The key insight is to allow the simple logit-based transferable attacks to converge. Extensive results presented well supported the claims.

**Limitations And Societal Impact:**

No Limitations and societal impact.

**Main Review:**

Strength:
- The finding is very interesting - Compared to SOTA resource-intensive transferable attacks, simple logit-based transferable attacks can get stronger transferable adversarial examples without any additional training data and models.
- The paper is well written and easy to follow.
- Comprehensive experiments are presented. The improvement over Po+Trip is surprisingly good under some settings.

Weakness:
- The method used is extremely simple and not that novel, but this is somewhat acceptable since the interesting findings do greatly reduce the complexity of transferable attacks.


**Time Spent Reviewing:**

2

---

> ### Author Response · Authors · 2021-08-10
> **Response to Reviewer nyxN**
>
> Thank you for recognizing the value of our paper, especially the surprising aspects of our results and the degree to which we are able to reduce the complexity of transferable attacks.
> Descriptions of limitations and societal impact can actually be found in the last paragraph of our paper.

---

### Official Review · Reviewer_dgjq · 2021-07-16

**Rating:** 4
**Confidence:** 3

**Summary:**

In this work, authors point out that using more training iterations for targeted attack significantly improves transferability and propose a logit loss which shows competitive performance compared to other losses. Additionally, authors employ the proposed logit loss to craft a new data-free and training-free UAP.

**Ethics Review Area:**

["I don’t know"]

**Limitations And Societal Impact:**

I suggest authors provide a (theoretical) analysis for the effectiveness of logit loss over other losses and why increasing iterations help to improve the transferability of targeted attack.

**Main Review:**

Originality: Although it is intuitively expected that with increasing iterations the attack becomes more effective, to my knowledge, it was not shown before for transferability of targeted attacks.
Similar losses, however, were already pointed out in previous works (for instance [1]), so the idea of the logit loss is not very novel.

[1] Naseer, M. M., Khan, S. H., Khan, M. H., Shahbaz Khan, F., & Porikli, F. (2019). Cross-domain transferability of adversarial perturbations. Advances in Neural Information Processing Systems, 32, 12905-12915.

Quality: Claims are well supported by experiments.

Clarity: Paper is well written, it is easy to follow and understand.

Significance: Results do not look significant to me since most of them are expected and/or were mentioned in previous works.

Strengths: The main ideas of the paper are simple, easy to understand and implement. The claims are well supported empirically.

Weaknesses: However, the main contributions of the paper are not novel enough. The phenomenon that increasing the number of iterations improves the transferability of the targeted attack does not look surprising to me, although admittedly it was not explicitly stated before. As for the proposed loss, similar losses were suggested by previous works, so the idea of the logit loss is not new. In addition, no theoretical analysis or guarantees were provided as to why it is better to train for more iterations to achieve better transferable targeted attack or why the logit loss outperforms other losses.


**Time Spent Reviewing:**

4

---

> ### Author Response · Authors · 2021-08-10
> **Response to Reviewer dgjq**
>
> **Q1:** The idea of the logit loss is not novel because similar losses were already pointed out by [1].
>
> **A1:** Our technical contribution is that we identify the superiority of the simple logit loss for targeted transferability, but not (as we explicitly acknowledge in the paper) the technical novelty of the loss itself. In our paper, we have conducted detailed comparisons of the logit loss to the most relevant losses (CE and Po+Trip) for our task and also demonstrate that its performance is superior to that of the most similar loss, the C&W loss. The authors of [1] have also acknowledged that their adopted Relativistic Cross-Entropy (RCE) loss is not new, but a well-known loss for training GANs. Despite its limited technical novelty, [1] is still valuable work because of their insightful observations that the RCE loss is superior in the context of training GANs for generating transferable non-targeted perturbations.
>
> On the other hand, we respectfully disagree that the RCE loss in [1] and our logit loss are similar. Also, we do not think [1] is closer to our work than the studies we have compared in our paper. First, the RCE is a variant of CE loss and is still based on softmax probability. In contrast, the logit loss completely gets rid of the softmax. Second, these two losses are used in different tasks. Specifically, the RCE loss is used to train a GAN model over large-scale additional image data for generating perturbations but our logit loss is used to directly optimize perturbations on a per-image basis. Third, [1] is focused on non-targeted transferability and no experiments on targeted transferability have been conducted. In contrast, we focus our study on targeted transferability, which is well known to be much more challenging and has different properties from non-targeted transferability.
>
> **Q2:** Results do not look significant to me since most of them are expected and/or were mentioned in previous works.
>
> **A2:** We respectfully disagree that our results are expected since until now research on transferable targeted attacks has failed to reflect an awareness of our findings or methodological recommendations. We are a bit confused by the statement that the results were mentioned in previous work. The review does not provide references and later the reviewer states “although admittedly it was not explicitly stated before.”
>
> **Q3:** Theoretical analysis to explain the superiority of the logit attack.
>
> **A3:** Please see our response to Q1 of the Reviewer yhEq as shown above.
>
> **Q4:** Theoretical analysis to explain why using more iterations is helpful.
>
> **A4:** The reason is that the attacks can converge to their optimal targeted transferability when we do not unreasonably restrict them to few iterations, as is done in current research. This reason is stated explicitly in our paper and is well validated by our extensive experiments. Our theoretical analysis requested in Q3 has also explained why logit attack is especially effective when using more iterations.

---

### Official Review · Reviewer_yhEq · 2021-07-20

**Rating:** 6
**Confidence:** 3

**Summary:**

This paper suggests to use logit loss to replace the cross-entropy loss for adversarial attack, and shows that logit loss with more iterations can lead to more robust transferable attacks, outperforming TTP and FDA. Extensive experiments verify the proposed method.

**Ethics Review Area:**

["I don’t know"]

**Limitations And Societal Impact:**

No discussion.

**Main Review:**

My major concern of the paper is that the proposed method is not novel  --- the eq (9) seems so naive that I don't understand why it works so very. I hope the author could give more insights, or use a toy example to demonstrate the insights behind the method.

I also wonder whether we can combine eq (9) with some previous methods such as FDA and TTP, to get more robust attacks.

At last, I hope the author could share their code, including the baselines on ImageNet and Google Cloud Vision APIs. That would help the audience to reproduce the results and test the proposed method in different scenarios.

**Time Spent Reviewing:**

2 hours

---

> ### Author Response · Authors · 2021-08-10
> **Response to Reviewer yhEq**
>
> **Q1:** Theoretical analysis to explain the superiority of the logit attack.
>
> **A1:** The task of generating perturbations via optimization with many iterations on a per-image basis is different from normal training tasks that are aimed at generalization on unseen data via optimization over large-scale training data. In our case, as we have stated in our paper, the commonly used Cross-Entropy (CE) loss will make the gradient decrease and tend to vanish as we increase the number of iterations. This will prevent the attack optimization from achieving strong effects. A detailed, theoretical analysis of this problem is provided in [21], which we cited. However, we did not repeat the analysis in our paper. In the following text, we will present the theoretical analysis of this problem of CE, and also explain why the logit loss can solve this problem and also solve it in a better way than the Po+Trip loss proposed by [21]. We would also like to add this new analysis to our paper.
>
> As can be observed from Equation (1), the gradient of CE loss w.r.t the target logit input, $z_{t}$, will monotonically decrease as the target probability, $p_{t}$, increases during the attack optimization. In addition, due to the use of the softmax function, $p_{t}$ will quickly reach 1, and as a result, the gradient tends to vanish. This phenomenon makes the attack hard to improve even with more iterations. In contrast, the gradient of the logit loss, as shown by Equation (2), equals a constant. As a result, the attack can keep improving as we increase the number of iterations.
>
> $L_{CE} = -1 \cdot \log(p_{t}) = -\log({\frac{e^{z_{t}}}{\sum e^{z_{j}}}}) = -z_{t} + \log(\sum e^{z_{j}})$
>
> $\frac{\partial L_{CE}}{\partial z_{t}} = -1 + \frac{\partial \log(\sum e^{z_{j}}) }{\partial e^{z_{t}}} \cdot \frac{\partial e^{z_{t}}}{\partial z_{t}} = -1 + {\frac{e^{z_{t}}}{\sum e^{z_{j}}}} = -1 + p_{t}$ &nbsp;&nbsp;&nbsp;&nbsp;    (1)
>
> $L_{Logit} = -z_{t}$
>
> $\frac{\partial L_{Logit}}{\partial z_{t}} = -1$  &nbsp;&nbsp;&nbsp;&nbsp;(2)
>
>
> Although Po+Trip [21] has also addressed the problem of CE, it has taken a very aggressive strategy. Specifically, it arbitrarily reverses the growth of the gradient, i.e., making the magnitude of the gradient gradually increase during the optimization. However, we found that this arbitrary reversal of gradients has introduced negative effects since it causes the gradient descent-based optimization to overshoot the minima. As a result, the loss can not be minimized effectively. This issue of Po+Trip also explains why Po+Trip sometimes performs even worse than CE, especially in the case that involves non-smooth loss surfaces, e.g. the ensemble transfer setting with little model similarity (as shown in our Figure 3).
>
> To better illustrate the superior performance of the logit attack over others, we follow [1,21] to plot the loss and gradient trends, and also the target logit trend over iterations. For normalization, we rescale all the losses/gradients by dividing their values of the first iteration. We use the L1 (Taxicab) norm to represent the gradient magnitude. As can be observed by the figures shown in https://www.dropbox.com/s/vdo0w566ahy5lr4/trends.PNG?dl=0, the loss value of CE quickly approaches zero and the Po+Trip loss stays almost unchanged with a small oscillation. In contrast, the logit loss can be minimized continuously over iterations with relatively large gradients. Overall, the use of the logit loss makes sure that the target logit is continuously increased over iterations and finally reaches a very high value.
>
> [1] Naseer et al., Cross-domain transferability of adversarial perturbations, NeurIPS 2019.
>
>
> **Q2:** Integrate logit loss with previous methods such as FDA and TTP.
>
> **A2:** It would be interesting to see if the logit loss could help improve resource-intensive methods, such as FDA and TTP. For FDA, it is clear that we can directly replace the original attack loss with the logit loss since FDA is also an iterative attack. Unfortunately, since the code and pre-trained models of FDA have not (yet) been released, we are not able to explore it at this stage but would leave it for future work.
> For TTP, which is based on training a GAN generator, it might be possible to use the logit loss as (part of) the training loss function. This is interesting and also challenging to try since typical model training commonly relies on class probability (e.g. CE loss) rather than directly using the logit output. We would like to explore this in the future should the codes for training the GANs in TTP be publicly available.
>
> **Q3:** Open source the code.
>
> **A3:** We did submit the source code of our experiments on ImageNet as the supplementary material. We will make all the code (including that for Google Cloud Vision experiments) publicly available should the paper be accepted.

---

### Author Response · Authors · 2021-08-10
**Response to all reviewers regarding our contributions.**

Our work opens a new way forward for transferable targeted attacks by establishing that success can be achieved through simplicity (naive logit loss) given the right methodological framework (using more iterations). Until now, this point has been overlooked in the literature. Novel attacks such as FDA and TTP are of interest, but the importance of such technical innovations must be seen in the proper light, namely, relative to the corrected understanding of previously existing approaches which is offered by our paper. Our work will improve the validity and meaningfulness of research on transferable targeted attacks as reflected by the reviewer who remarks on the usefulness of the findings for researchers in the field. In focusing on addressing methodological issues with simple (existing) techniques, we pursue a similar aim as recent influential work (representative examples are [1-6]).


[1] Carlini and Wagner, Adversarial examples are not easily detected: Bypassing ten detection methods, AISec 2017.

[2] Athalye et al., Obfuscated gradients give a false sense of security: Circumventing defenses to adversarial examples, ICML 2018.

[3] Wong et al., Fast is better than free: Revisiting adversarial training, ICLR 2020.

[4] Rice et al., Overfitting in adversarially robust deep learning, ICML 2020.

[5] Tramèr et al., On adaptive attacks to adversarial example defenses, NeurIPS 2020.

[6] Pang et al., Bag of tricks for adversarial training, ICLR 2021.

---

### Decision · Program_Chairs · 2021-09-28

**Decision:**

Accept (Poster)

**Comment:**

This paper studies targeted attacks, where one of the main findings is showcasing that the simple logit loss can lead to very good transferability of attacks. The reviewers liked the contributions of the paper, partially because the logit loss and associated approach was simple but effective. The reviewers also appreciated the experiments in this paper and found them interesting and well executed. In light of all of this, I recommend acceptance.

The reviewers do also point out a couple of related works, such as [1] below, and I encourage the authors to discuss their findings in the context of this work and other prior work. The authors have also made some promises, such as to open source their code, and I would encourage them to follow through on any updates that have been pointed out in the reviews.

[1] Chaoning Zhang, Philipp Benz, Tooba Imtiaz, and In So Kweon. Understanding adversarial examples from the mutual influence of images and perturbations. In CVPR, 2020.

**Consistency Experiment:**

NeurIPS has a long history of experimentation. In 2014, NeurIPS ran an experiment in which 10% of submissions were reviewed by two independent committees to quantify the randomness in the review process. This year, we repeated a variant of this experiment to see how the quality of the review process has changed over time.  This paper was part of the experiment and was therefore assigned to two committees (consisting of reviewers, an Area Chair, and a Senior Area Chair) that reached independent decisions.  If both committees made the same recommendation, this recommendation was followed. If a single committee recommended acceptance, the paper was accepted (with the exception of a few cases in which the other committee identified what we considered a fatal flaw, e.g., an error in a key result).

This copy’s committee reached the following decision: **Accept (Poster)**

The other committee assigned to the paper recommended **Reject**.  You can find the other set of reviews, along with any follow up discussion with the authors here:
https://openreview.net/forum?id=aHK-onEhYRg